# Improved Antibody Detection for Canine Leptospirosis: ELISAs Modified Using Local Leptospiral Serovar Isolates from Asymptomatic Dogs

**DOI:** 10.3390/ani14060893

**Published:** 2024-03-14

**Authors:** Pannawich Boonciew, Watcharee Saisongkorh, Suppalak Brameld, Matsaya Thongpin, Alongkorn Kurilung, Pratomporn Krangvichian, Waree Niyomtham, Kanitha Patarakul, Thanmaporn Phichitraslip, David J. Hampson, Nuvee Prapasarakul

**Affiliations:** 1Department of Veterinary Microbiology, Faculty of Veterinary Science, Chulalongkorn University, Henri-Dunent Street, Pathumwan, Bangkok 10330, Thailand; pannawichb12@gmail.com (P.B.); pewwaree@yahoo.com (W.N.); 2Laboratory of Immunology, National Institute of Health, Department of Medical Sciences, Ministry of Public Health, Nonthaburi 11000, Thailand; watcharee.s@dmsc.mail.go.th (W.S.); suppalak37@gmail.com (S.B.); matsaya.t@dmsc.mail.go.th (M.T.); 3Siriraj Metabolomics and Phenomics Center, Faculty of Medicine Siriraj Hospital, Mahidol University, Bangkok 10700, Thailand; alongkornkurilung@gmail.com; 4Medical Microbiology, Interdisciplinary Program, Graduate School, Chulalongkorn University, Bangkok 10330, Thailand; p_krangvichian@hotmail.com; 5Department of Microbiology, Faculty of Medicine, Chulalongkorn University, Bangkok 10330, Thailand; kanitha.pa@chula.ac.th; 6Chula Vaccine Research Center (Chula VRC), Center of Excellence in Vaccine Research and Development, Chulalongkorn University, Bangkok 10330, Thailand; 7Faculty of Veterinary Technology, Kasetsart University, Bangkok 10900, Thailand; cvttpp@ku.ac.th; 8School of Veterinary Medicine, Murdoch University, Perth, WA 6150, Australia; d.hampson@murdoch.edu.au; 9Center of Excellence in Diagnostic and Monitoring of Animal Pathogens (DMAP), Chulalongkorn University, Bangkok 10330, Thailand

**Keywords:** canine leptospirosis, diagnosis, serological test, microscopic agglutination test, enzyme-linked immunosorbent assay, local serovars or isolates

## Abstract

**Simple Summary:**

Leptospirosis is an infectious zoonotic disease caused by pathogenic Leptospira and affects both humans and animals throughout the world. Domestic animals, such as dogs, can act as potential reservoirs of leptospires for human or animal infections and environmental contamination. Due to the different varieties of the predominant local circulating pathogenic serovars in several regions, serodiagnosis tests, such as conventional microscopic agglutination tests (MATs) or others that do not include local serovars and their protein antigens, have limitations, and may fail to detect the disease and carry out antibody surveillance in certain regions. This study aimed to develop a more accurate antibody detection tool for canine leptospirosis in our region by using an indirect enzyme-linked immunosorbent assay (ELISA) using relevant local isolates of leptospiral serovars from asymptomatic dogs. All the modified IgG-ELISAs with the local isolates gave positive results for all infected dogs, especially outer membrane protein (OMP)-based IgG-ELISAs that showed negative results for all dogs from non-endemic areas, demonstrating improved accuracy and reduced limitations over those of the standard MAT and providing an enhanced method for leptospirosis detection in the study area. This improvement is crucial when investigating the epidemiology of the disease and preventing its spread. The article highlights the requirement and significance of utilising local circulating isolates in serological approaches to accurately diagnose and monitor leptospirosis.

**Abstract:**

Leptospirosis is a zoonotic disease of significant concern for human and animal health, with domestic animals, including dogs, acting as reservoirs for human infection. Serology is widely used for leptospirosis diagnosis, even though the standard microscopic agglutination test (MAT) using a panel of serovars lacks specificity and can lead to detection limitations in certain regions. In this study, we aimed to develop an antibody detection tool for dogs using an indirect enzyme-linked immunosorbent assay (ELISA) with a set of local serovar isolates, including Paidjan, Dadas, and Mini, to enhance the accuracy of leptospirosis surveillance in our region. The specificity and sensitivity of various antigen preparations, namely leptospiral whole-cell protein (WCP), total membrane protein (TMP), and outer membrane protein (OMP), were assessed using sera from infected and non-infected dogs, as well as negative puppy sera. Leptospirosis diagnosis was supported using a genus-specific nested polymerase chain reaction test on all collected sera. Protein preparations were validated using SDS-PAGE and Western blotting analysis. In the results, the standard MAT failed to detect antibodies in any of the dogs confirmed as being infected using PCR and isolation, highlighting its limitations. In contrast, the OMP-based ELISAs using local isolates of *Leptospira* serovars gave positive results with sera from all infected dogs, and negative results with sera from all dogs from non-endemic areas. IgG titres of infected and unvaccinated dogs from endemically affected areas were significantly higher than those in non-endemic regions. Using the OMP-based IgG/ELISAs with the local serovar Dadas resulted in higher specificity and lower sensitivity than when using the WCP- and TMP-based IgG/ELISAs. Agreement analysis revealed fair and moderate concordance between OMP-based IgG/ELISAs and PCR results, whereas slight and fair agreement was observed between OMP-based ELISAs and the MAT. Overall, the modified OMP-based IgG/ELISAs, utilising relevant local serovar isolates from dogs, demonstrated improved accuracy in detecting leptospirosis in the study area, overcoming the limitations of the MAT. This study highlights the importance of identifying and incorporating these local circulating serovar isolates into serological techniques for leptospirosis diagnosis and surveillance.

## 1. Introduction

Leptospirosis is considered a worldwide significant infectious zoonosis for humans and animals, particularly in developing countries, including Thailand [1,2,3,4]. The disease is caused by pathogenic spirochetes in the genus *Leptospira*, which currently contains 68 species, in which pathogenic groups can be divided into over 24 pathogenic serogroups and 250 pathogenic serovars based on the surface epitopes of the lipopolysaccharide (LPS) antigens on their outer membrane [2,5,6,7]. As various pathogenic serovars can infect humans and animals, it is crucial to identify the serovar involved and develop improved serological diagnostic techniques to understand their epidemiology and prevent the spread of disease [5,8,9,10,11,12,13].

Mammals, especially domestic animals (dogs and livestock), can play a significant role as reservoirs for disease maintenance and transmission, contaminating the environment via their urine [1,14]. The four pathogenic serovars Canicola, Icterohaemorrhagiae, Grippotyphosa, and Pomona can be found in dogs, as assessed by antibody prevalence or isolation, and have been included in several commercial canine vaccines and diagnostic methods [15,16]. Despite this, studies on antibody prevalence have suggested that the serovars circulating in dogs vary among locations.

According to studies previously conducted in Thailand, various leptospiral serovars, including Autumnalis, Australis, Bataviae, Bratislava, Canicola, Copenhageni, Grippotyphosa, Icterohaemorrhagiae, Javanica, Mini, Pomona, Sejroe, Shermani, and Tarassovi have been shown to have high seroprevalence in dogs; moreover, these serovars also have been described as the predominant circulating serovars found in humans in Thailand [17,18,19,20,21]. Previous reports demonstrate that dogs may spread and increase the probability of human infection with the circulating serovars, where dogs and humans share the same areas and environment [22,23,24,25].

In Thailand, *L. interrogans* serovars Paidjan, Dadas, and Batavaie, and *L. weilii* serogroup Mini were the leptospires most frequently detected in asymptomatic dogs via a cross-agglutination test, whereas evidence for their presence could not be detected via the MAT. Phylogenetic analysis indicated that they were genetically related to *L. interrogans* isolates from the urine of asymptomatic humans with a previous history of symptoms and close contact with unvaccinated animals in their backyards (*L. interrogans* serovars Paidjan strain CUDO5 and Dadas strain CUDO8) [26,27]. In addition, two local canine isolates of *L. weilii* belonged to sequence type (ST) 94 (containing an undesignated serovar and strains CUDO6 and CUDO13), and this ST was closely related to ST183 (containing an undesignated serovar and strain LNT1234) and ST193 (containing serovar Hekou, strain H27 of *L. weilii*), isolated from humans in Laos and China [26,27,28,29,30]. Moreover, these serovars, especially Bataviae, have been reported to cause disease in humans and dogs and are carried by rodents [13,31,32,33,34]. Overall, dogs represent a significant potential source of human leptospirosis infection [26,32,35].

Current methods for diagnosing animal leptospirosis follow recommendations by the World Organisation for Animal Health (OIE), focusing either on detecting the agent or immune response indicating past exposure [36,37,38]. PCR is the gold-standard method for detecting early and chronic infection in agent detection, while serum antibody detection using the MAT with live leptospires or an ELISAs with protein antigens is suggested for surveillance [38,39]. The serovars used in the test should reflect the serovars circulating in the investigated populations [36,37].

Recent studies in Thailand have identified issues with the use of serological and PCR detection methods for guiding the country’s strategic approach to leptospirosis. The local Leptospiral isolates, specifically serovars Paidjan and Dadas for *L. interrogans* and serogroup Mini for *L. weilii*, are not included in the 24 serovars used in the MAT panel by Thailand’s National Institute of Health. The MAT panel includes 23 pathogenic serovars and 1 non-pathogenic serovar [40,41]. Challenges arise due to differences in strains and serovars, impacting diagnostic accuracy. Discrepancies between PCR and MAT results have been observed, impacting the application of serological detection for surveillance in Thailand [26]. To address these limitations, several studies have developed ELISAs to detect IgM and IgG antibodies against leptospiral antigen proteins, which are cost-effective, offer good specificity and sensitivity, and are suitable for the large-scale surveillance of animal populations [42,43,44].

This study aimed to assess the effectiveness of using locally sourced leptospiral isolates from dogs in Thailand for modified indirect ELISAs by comparing these with the standard MAT test, which utilises conventional serovars, to enhance the accuracy of leptospirosis detection in canine sera. Different protein antigen preparations comprising whole-cell, total membrane, and outer membrane protein were evaluated to provide recommendations for optimal antigen selection to improve test sensitivity and specificity.

## 2. Materials and Methods

### 2.1. Bacterial Cultures and Growth Conditions

Nine *Leptospira* isolates were used for antibody detection in the MAT and for protein extraction in developing modified ELISAs. These comprised five local leptospiral isolates that were found in the urine of asymptomatic dogs in Thailand, including three from *Leptospira interrogans*: serogroup Bataviae, serovar Paidjan, strain CUDO5; serogroup Grippotyphosa, serovar Dadas, strain CUDO8; and serogroup Bataviaem serovar Bataviae, strain D64. They also comprised two *Leptospira weilii* strains: serogroup Mini strains CUDO6 and CUD13 (non-identified serovars) [26,32], and four isolates of *L. interrogans* that are commonly used in commercial leptospirosis vaccines in dogs, comprising serovars Icterohaemorrhagiae, Pomona, Grippotyphosa, and Canicola. The methods used to control biohazards associated with handling Leptospira were approved by the Institutional Biosafety Committee of the Faculty of Veterinary Science, Chulalongkorn University (IBC 2031004). All leptospiral isolates were cultivated and incubated in 25 mL of liquid Ellinghausen–McCullough–Johnson–Harris (EMJH) medium (Difco, Sparks, MD, USA) supplemented with enrichment EMJH (Difco, USA) and 3% (*v*/*v*) rabbit serum for two weeks in aerobic conditions at 28–30 °C. The other leptospiral standard serovars used in the MAT panel were provided by the National Institute of Health (NIH), Thailand [40,41]. The list of all 29 isolates of 27 serovars used is shown in Table 1.

### 2.2. Serum Samples and Groups of Dogs

Whole canine sera were collected from 260 dogs and divided into five groups based on their status and origin. The collection of serum samples from dogs was approved by the Chulalongkorn University Animal Care and Use Committee (CU-ACUC; Protocol NO. 1531075, groups 1 and 2). Moreover, the remaining canine sera from another study were authorised by the owners with written informed consent for the participation of their dogs in this study (groups 3, 4, and 5). Group 1 (n = 6) included leptospiral-infected dogs in an area endemic to local isolates of leptospiral serovars (Nan Province) from a previous study. Leptospirosis infection was confirmed by testing urine using a nested PCR targeting a genus-specific region on the leptospiral 16S ribosomal RNA (*rrs*) gene. The four leptospiral serovar isolates were also obtained from the urine of these dogs [26]. The serogroups and serovars of these isolates were identified by using the microscopic agglutination test (MAT) with polyclonal and monoclonal antibodies raised against *Leptospira* isolates at the OIE National Collaborating Centre for Reference and Research on Leptospirosis, The Netherlands [27,30]. The isolates were identified as *L. interrogans* serovars Paidjan and Dadas and *L. weilii* serogroup Mini, as mentioned earlier. Group 2 (n = 21) included unvaccinated dogs from the same endemic area as that of the positive leptospiral-infected dog group from a previous study [26]. The urines of these dogs were all negative for *Leptospira*, as determined via nested PCR and isolation. Group 3 (n = 112) was a set of one-year-old dogs from Bangkok with a history of a complete vaccination program from the Blood Bank Unit, Small Animal Hospital, Faculty of Veterinary Science, Chulalongkorn University. Each dog had received various combinations of vaccine isolates from the multivalent leptospiral vaccine (bivalent, trivalent, or quadrivalent leptospiral vaccine) when they were vaccinated at different hospitals or small animal clinics before blood donation. The quadrivalent vaccine for leptospirosis contained four leptospiral serovars from the species *Leptospira interrogans*, including serovars Icterohaemorrhagiae, Canicola, Grippotyphosa, and Pomona (trivalent vaccine: serovars Icterohaemorrhagiae, Canicola, and Grippotyphosa; bivalent vaccine: serovars Icterohaemorrhagiae and Canicola). Group 4 (n = 108) comprised unvaccinated dogs from non-endemic areas for the isolated serovars (Bangkok, Samut Prakan, and Chonburi provinces). Group 5 (n = 13) included two-month-old unvaccinated puppies from the same non-endemic areas as those represented by Group 4, and these served as a negative control to determine the cut-off values. The sera from Groups 4 and 5 were provided by staff of the Faculty of Veterinary Technology, Kasetsart University. All the canine sera were tested for the *Leptospira* genus gene via nested PCR to help investigate any history of leptospirosis infection.

### 2.3. Detection of Leptospira in Sera Using a Genus-Specific Nested PCR Assay

DNA was extracted from 100 μL samples of each canine serum using Genomic DNA Extraction and Purification Kit (Thermo Scientific™, Thermo Fisher Scientific, Waltham, MA, USA), following the manufacturer’s instructions. DNA extracts were stored at −20 °C before use. A single-tube nested PCR was used to detect a genus-specific region on the leptospiral 16S ribosomal RNA (*rrs*) gene found in pathogenic and intermediate pathogenic *Leptospira* spp., as previously described [45]. A 25 μL nested PCR was performed using two sets of primers, including rrs-outer-F (5′-CTCAGAACTAACGCTGGCGGCGCG-3′), rrs-outer-R (5′-GGTTCGTTACTGAGGGTTAAAACCCCC-3′), rrs-inner-F (5′-CTGGCGGCGCGTCTTA-3′), and rrs-inner-R (5′-GTTTTCACACCTGACTTACA-3′). The sizes of the final amplicon were 547 and 443 bp. The genomic DNA extracted from the local isolates of the leptospiral serovars from *L. interrogans* and *L. weilii* was used as the positive control. In contrast, the negative control was the reaction mixture without a DNA template. Both positive and negative controls were included in each run of the nested PCR assay. The resulting amplicons were separated via 1.5% agarose gel electrophoresis and visualised under ultra-violet light following staining with ethidium bromide. The DNA extracts of a serum sample showing bands of resulting amplicons were recorded as positive PCR results.

### 2.4. A Preliminary Study Using the Microscopic Agglutination Test (MAT)

Fifty serum samples including all of those in Group 1 (6 sera) and Group 2 (21 sera), and a subset of 23 serum samples from Group 3 were selected for use in a preliminary study of the MAT. This was conducted to test the hypothesis that the use of the approved standard MAT might not detect antibody titres in the sera samples from dog groups in this study, especially leptospiral-infected dogs. In the MAT panel, 29 isolates of 27 serovars were used, including 24 representative reference isolates of *Leptospira* serovars [40,41] and 5 local isolates of *Leptospira* serovars isolated from Thai asymptomatic dogs [26,32]. The MAT procedure was carried out as previously described [46]. Individual serum samples were tested via two-fold dilutions from 1:20 to 1:10,240. The threshold of MAT cut-off for reactivity was defined as ≥1:20 [32]. The titre, the maximum dilution at which 50% of leptospires agglutinated with the antibody from the serum dilution, was used to interpret the MAT results.

### 2.5. Leptospiral Protein Preparations for Modified ELISAs

#### 2.5.1. Whole-Cell Protein Using Sonicated Leptospiral Preparations

All leptospiral protein preparations were extracted in duplicate using 50 mL of *Leptospira* EMJH culture (one replicate; 25 mL). After protein preparations were completed, the extracted proteins from each replicate were pooled in a single tube. Whole-cell protein (WCP) extraction was modified from a previously described method [47]. In total, of 2.1–2.9 × 10^8^ cells/mL of each *Leptospira* isolate cultured in EMJH medium was washed three times with phosphate-buffered saline (PBS, pH 7.4) and centrifuged at 13,000× *g* at 4 °C for 10 min. The cells were subjected to 10 cycles of −80 °C freezing and 10 min of thawing at room temperature. Each *Leptospira* preparation was then sonicated for 30 min at 4 °C in an ice bath. Cells containing PBS were mixed using SiLibeads Type S with an equivalent volume of PBS, and ten cycles of shaking for 1 min and cooling for 2 min were conducted. The WCP extracts were then centrifuged at 10,000× *g* for 5 min to elute them, after which each was aliquoted and kept at −20 °C until needed. The protein concentrations of the extracts ranged from 502 to 641 µg/mL for the five local isolates and 707 to 872 µg/mL for the four vaccine isolates of *Leptospira* serovars.

#### 2.5.2. Total Membrane Protein Fraction Obtained Using Lysis Buffer

Total membrane protein (TMP) extraction utilising lysis buffer was modified from a previously described method [48]. For each isolate, 2.1–2.9 × 10^8^ *Leptospira* cells cultured in EMJH medium were centrifuged three times at 13,000× *g* for 10 min at 4 °C using PBS for washing. The final *Leptospira* pellets were dissolved in 1 mL of lysis buffer (20 mM Tris (pH 8), 150 mM NaCl, 2 mM EDTA, and 2 mg of lysozyme per mL). Each live *Leptospira* serovar suspension was then sonicated after standing for 30 min in an ice bath at 4 °C. Finally, the *Leptospira* suspension was centrifuged at 13,000× *g* for 10 min at 4 °C to separate the complete membrane protein pellet from the soluble protein supernatant. For the ELISAs and protein concentration measurements, the TMP pellets were resuspended in PBS and stored at −20 °C. The concentrations of the TMPs for the five local isolates were 308 to 429 µg/mL, and these were 534 to 663 µg/mL for the four vaccine isolates of *Leptospira* serovars.

#### 2.5.3. Outer Membrane Protein Isolation Using Triton X-114

Outer membrane protein (OMP) extraction using Triton X-114 was modified from previously described methods [48,49,50,51]. Leptospiral isolates containing 2.1–2.9 × 10^8^ cells/mL in EMJH medium were washed in triplicate with PBS via centrifugation at 13,000× *g* and 4 °C for 10 min. Each *Leptospira* pellet was immersed in Triton X-114 buffer (20 mM Tris (pH 8), 150 mM NaCl, 2 mM EDTA, and 2% Triton X-114) for 1–2 h at 4 °C to extract the OMPs. The suspension was centrifuged at 17,000× *g* for 45 min to remove the insoluble pellet and preserve the supernatant. Phase separation was then conducted on the supernatant by adding PBS containing 20 mM CaCl_2_, subjecting it to warming at 37 °C for 1 h, and centrifuging it at 6000× *g* for 10 min at 25 °C. The protein phase of each *Leptospira* isolate was divided into three phases: an aqueous phase, a detergent phase that contained outer membrane proteins, and pellets. The detergent phase was dissolved in PBS and stored at −20 °C for use in the ELISAs and measurements of protein concentration. The protein concentrations of all the isolates ranged from 113 to 227 g/mL.

### 2.6. Confirmation of Protein Components via SDS-PAGE and Western Blotting

SDS-PAGE and Western blotting techniques were modified from the methods described in previous studies [48,52]. The protein components of the three extractions from the five local isolates of *Leptospira* serovars were visualised via 12.5% polyacrylamide gel electrophoresis and staining with Coomassie Brilliant Blue R-250. For Western blotting, proteins in the polyacrylamide gels were transferred onto polyvinylidene difluoride (PVDF) membranes, which were equilibrated using absolute methanol. After protein transfer, the membranes were blocked with 5% skim milk in TBS buffer with 0.1% Tween (TBST) and washed three times with TBST after being incubated for 2 h at room temperature. Each membrane was incubated at 4 °C overnight with pooled leptospirosis sera from dogs confirmed as infected from Nan Province (Group 1) diluted 1:200 in TBST containing 5% Bovine serum albumin (TBST-BSA). The membranes were washed with TBST and incubated with 1:1000 horseradish peroxidase (HRP)-conjugated goat anti-dog IgG antibody in TBST at room temperature for 2 h. After incubation, the membranes were washed with TBST and visualised via chromogenic detection using a 3,3′-diaminobenzidine tetrahydrochloride (DAB) substrate.

### 2.7. Indirect Immunoglobulin G (IgG) ELISAs

#### 2.7.1. Optimisation of Indirect ELISAs and Cut-Off OD Values

The optimum single-working protein concentrations and serum dilutions were established using checkerboard titration with the indirect ELISA technique, modified according to previously described studies [53,54]. The protein obtained from each extraction technique and isolate was suspended in 2 µg/mL of 0.05 M carbonate-bicarbonate buffer (pH 9.6), serially diluted two-fold to a final dilution of 0.015625 µg/mL, and applied to the wells of 96 well-microplates with a total volume of 100 μL of diluted protein in coating buffer. Three different groups of dog sera were used to optimise the indirect ELISAs, including (1) pooled serum samples from six asymptomatic dogs with *Leptospira* successfully isolated from their urine or who tested antigen-positive via PCR (Group 1), (2) pooled serum samples from six vaccinated dogs older than one year and with a history of complete vaccination (Group 3), and (3) pooled serum samples from 13 non-vaccinated 2-month-old dogs without a history of vaccination (Group 5). Each pooled sera preparation was added to PBS buffer with 0.05% Tween 20 containing 1% Bovine serum albumin (PBST-BSA) at a starting dilution of 1:80,serially diluted two-fold to a final dilution of 1:81,920 and then added to the wells across the columns of the microplates. After the optimisation of protein concentrations and serum dilutions, all 13 sera from the unvaccinated 2-month-old dogs (Group 5) were individually used to detect the optical density (OD) value and define the cut-off OD value for each modified indirect ELISA using the optimal protein concentration and serum dilution with the same conditions as those of the indirect ELISA technique used before. Each individual serum sample was examined for the OD value in triplicate. All OD values from 13 sera in triplicate were then computed as the single optimal OD cut-offs using the sum of the mean ODs plus four standard deviations (Mean + 4SD) to enhance the specificity of the modified ELISA tests for distinguishing between positive and negative results. The use of 4SD aimed to minimise false positives while maintaining sensitivity, as in other field studies [55,56,57,58].

#### 2.7.2. IgG Antibody Detection

The optimal protein concentration for all three extraction methods (1 µg/mL) in 0.05 M carbonate-bicarbonate buffer (pH 9.6) was used to coat the wells (100 μL/well) overnight at 4 °C. After rinsing them three times with PBS buffer (pH 7.4), the plates were blocked with PBST-BSA blocking solution (100 μL/well) for 1 h before being washed three times with PBS buffer. The optimal serum sample dilutions for WCP, TMP, and OMP preparations were used at 1:1280, 1:640, and 1:640 with PBST-BSA, respectively. In total, 260 serum samples from the 5 dog groups were used. Each diluted serum sample was added to the well of the microplate (100 μL/well) and incubated at 37 °C for 1 h. After protein and serum incubation, plates were washed three times with PBS buffer. The bound IgG antibody was detected using a 1:10,000 dilution of goat anti-dog IgG conjugated with HRP enzyme in PBST-BSA buffer (100 μL/well) and incubated for 1 h at 37 °C. The microplates were washed three times with PBS buffer before 3,3′,5,5′-Tetramethylbenzidine (TMB) substrate (50 μL/well) was added to the wells, and the plates were then left at room temperature for 5–10 min to allow the development of a blue-green colour. Then, 2M sulphuric acid (50 μL/well) was used to terminate the colour development reaction, under which the colour changed to yellow. The IgG levels in individual serum samples were examined in duplicate. The reactivity was assessed by measuring the OD value at a 450 nm wavelength with a microplate ELISA reader. The OD of the blank control with PBST-BSA without serum dilution was subtracted from the OD of each well.

### 2.8. Statistical Analysis

The effectiveness of the modified indirect ELISAs was preliminarily assessed. The results of the modified ELISAs, PCR, and MAT assays were compared by utilising the first 50 serum samples selected from dog groups 1, 2, and 3. The evaluation included calculating the values for sensitivity, specificity, positive predictive value (PPV), negative predictive value (NPV), intra-assay (repeatability), inter-assay (reproducibility), and accuracy. The OD values obtained via IgG detection for all 260 serum samples from all serum groups, including the first 50 serum samples from dog groups 1, 2, and 3, that were used to evaluate the diagnostic performance of the modified ELISAs were calculated and analysed using nonparametric statistics via the Kruskal–Wallis test and Dunn’s post hoc test in GraphPad prism software v.9 to compare the differences in the levels of IgG antibody between groups of dog sera within each modified indirect ELISA. A receiver–operator curve (ROC) analysis was determined and used to establish the appropriate cut-off OD values for the specificity and sensitivity of each modified ELISA test using MedCalc software v.20.006. The association between the antibody titre of the MAT and the antibody level of the modified ELISAs was determined using the 14 serum samples with MAT titres above a 1:20 dilution from dog groups 1, 2, and 3 via Pearson correlation analysis. Using agreement analysis and Cohen’s kappa statistics, the degree of concordance among the three diagnostic methods was determined by comparing the results for the 50 serum samples (groups 1, 2, and 3) with the modified ELISAs against PCR and the MAT assays.

## 3. Results

### 3.1. Detection of Leptospira in Serum via Nested PCR

The results of the nested PCR for the 260 serum samples are shown in Table 2. Only the six serum samples of infected dogs from the endemic area (Group 1) tested positive.

### 3.2. Antibody Titres in the Microscopic Agglutination Test

The 50 serum samples, which were selected from the three dog groups, Group 1, 2 and 3, were tested via the MAT against 29 isolates (27 serovars), including 24 representative reference isolates and 5 local isolates of *Leptospira* serovars. Table 2 and Table 3 record the number of evaluated sera and antibody titres against the serovars. Of the 50 serum samples, only 11 (11/21; 52%) sera from unvaccinated dogs in the endemic area (Group 2) and 3 (3/23; 13%) sera from vaccinated dogs in the non-endemic areas (Group 3) had antibody titres equal to or greater than a 1:20 dilution. In contrast, no sera (0%) from positive leptospiral-infected dogs in the endemic areas (Group 1) had antibody titres equal to or above the 1:20 dilution. The positive MAT serum samples from groups 2 and 3 were in the 1:20 to 1:80 dilution range, with Group 2 having higher antibody titres than group 3. In addition, antibody titres were only detected for four *Leptospira* serovars, including serovars Hebdonadis, Sejroe, Shermani, and Paidjan, with the numbers of positive sera to these serovars in MAT being four, one, nine, and one, respectively. One serum sample from an unvaccinated dog in the endemic area (Group 2) had antibody titres against the local isolate of serovar Paidjan, previously recovered from asymptomatic dogs. One serum sample from the same group displayed antibody titres with serovars Hebdonadis and Shermani, which are reference serovars used in standard MAT panels in Thailand.

### 3.3. The Protein Components Confirmed via SDS-PAGE and Western Blot Analysis Using Serum from a Dog with Leptospirosis

The protein components of the three extractions from the five local leptospiral serovar isolates were assessed via SDS-PAGE and Western blotting before use in modified ELISAs, and the results are shown in Appendix A. The proteins from the preparations of WCP, TMP, and OMP were shown to react to pooled sera from a dog confirmed as positive for leptospirosis via isolation and PCR, and displayed different staining intensities and patterns.

### 3.4. Determination of the Optimal OD Cut-Off Values for Indirect ELISAs

Based on the optimal protein concentration and serum dilution for each antigen protein preparation, all 39 OD values from 13 unvaccinated 2-month-old dogs (Group 5) from non-endemic areas that acted as a negative control due to their history of no vaccination and the fact that they were PCR-negative for the leptospiral 16S ribosomal RNA (rrs) gene were examined in triplicate and computed to define the optimal OD cut-off values using the sum of the mean ODs plus four standard deviations (Mean + 4SD). The cut-off OD values for the ELISA using WCP, TMP, and OMP at a protein concentration of 0.1 µg/well (1 µg/mL) and serum dilutions of 1:1280, 1:640, and 1:640 were 0.593, 0.816, and 0.898, respectively. Serum samples that displayed an OD value equal to or greater than the values of the optimal OD cut-off were considered positive in the modified ELISAs that used the leptospiral serovar isolates in this study.

### 3.5. Detection of Antibody Levels via the Modified ELISAs

For the 260 serum samples, the number and percentage that tested positive in the modified ELISAs using five local isolates of *Leptospira* serovars and four *Leptospira* serovars commonly used in the leptospirosis vaccine in dogs are shown in Table 3 and Appendix A. Sera from all infected dogs with positive infection status confirmed via urine PCR and isolation (group 1) had positive antibody results under all ELISAs modified using local isolates of *Leptospira* serovars. At the same time, they did not exhibit positive titres under the MAT or in the ELISAs using the common *Leptospira* serovars used in canine vaccines. However, some sera from all unvaccinated dogs from the concurrent endemic area (Group 2) and vaccinated dogs from Bangkok (Group 3) that were negative under serum PCR tested positive under the MAT and modified ELISAs. In the WCP and TMP-based ELISAs using local isolates of *Leptospira* serovars, 52% to 71% and 0% to 33% of sera from groups 2 and 3, respectively, were positive. Using the OMP-based ELISAs, 43 to 62% of the sera from Group 2 were positive, but none of the sera from Group 3 were positive. Moreover, all sera from unvaccinated dogs from non-endemic areas (Group 4) and unvaccinated puppies from non-endemic areas (Group 5) were negative in all ELISAs utilising local isolates of *Leptospira* serovars. On the other hand, the OMP-based ELISAs coated with the set of serovars used in the vaccine gave strongly positive results for the vaccinated dogs (Group 3), for 66% to 76% of the sera. In contrast, infected dogs (group 1) and unvaccinated dogs (Group 2) were negative. Additionally, some of the sera, 21% to 31%, from the unvaccinated dogs in Group 4 were positive, whereas none of the sera from Group 5 were positive in any modified ELISAs utilising the *Leptospira* serovars commonly used in the canine leptospirosis vaccine.

The distribution of IgG antibody levels against the OMP of five local isolates of *Leptospira* and four common *Leptospira* serovars used in vaccines in the modified ELISAs are shown in Figure 1, and Appendix A. All OMP-based ELISAs using local isolates of *Leptospira* serovars identified IgG antibodies in dogs from the endemic area (groups 1 and 2), with a highly significant difference (*p*-value < 0.05) from dogs in non-endemic areas (groups 3, 4, and 5). On the other hand, some of the WCP and TMP-based ELISAs using local isolates of *Leptospira* serovars did not significantly differentiate (*p*-value > 0.05) between the groups of dogs from the endemic and non-endemic areas (Figure 1, and Appendix A). In contrast, all the modified ELISAs using the common *Leptospira* serovars used in vaccines showed that only the IgG antibody levels of the vaccinated dogs from non-endemic areas (Group 3) were significantly higher (*p*-value < 0.05) than those in the four other dog groups from the endemic and non-endemic areas (Appendix A).

### 3.6. Diagnostic Performance of All Modified ELISAs

The sensitivity, specificity, and intra- and inter-assay comparisons for all the modified ELISAs are shown in Appendix A. The diagnostic performance of the modified ELISAs was calculated using the results of the serum PCR, MAT, and ELISAs with 50 sera from three dog groups (groups 1, 2, and 3). All the modified ELISAs had the highest sensitivity compared with PCRs from both urine and sera. At the same time, OMP-based ELISAs showed the highest specificity, followed by TMP-based ELISAs and WCP-based ELISAs. For the comparison of the ELISA with the MAT, the highest sensitivity was for WCP-based ELISAs, whereas the highest specificity was for OMP-based ELISAs. In comparing the sensitivity among all modified ELISAs to that of the MAT, all ELISAs from the serovar Dadas produced the highest sensitivity, at 64.3 to 85.7%. However, the highest specificity was mostly found with OMP-based ELISAs, mainly where serovar Mini06 and Bataviae were used. The intra-assay (repeatability) and inter-assay (reproducibility) values of all three ELISA protein platforms were assessed for diagnostic precision. The percentage of the coefficient of variation (% CV) determined via the intra-assay analysis of the three modified ELISA platforms were 6.3, 4.1, and 2.6 for WCP, TMP, and OMP-based ELISAs, respectively. Furthermore, the inter-assay assessment showed that the percentage of the coefficient of variation for the three ELISA platforms based on WCP, TMP, and OMP were 11.0, 9.3, and 8.5, respectively.

### 3.7. Receiver–Operator Curve (ROC) Analysis, Correlation Analysis, and Agreement Analysis

The receiver–operator curve (ROC) analysis and area under the ROC curve (AUC) for the determination of the cut-off OD value for the specificity and sensitivity of each modified ELISA are shown in Figure 2 and Appendix A. The AUC of the modified ELISAs against WCP/IgG-ELISAs, TMP/IgG-ELISAs, and OMP/IgG-ELISAs ranged from 0.610 to 0.676, 0.607 to 0.696, and 0.585 to 0.691, respectively. The cut-off OD values of each modified ELISA were 0.564–0.744, 0.413–0.865, and 0.2695–0.899 for the WCP/IgG-ELISAs, TMP/IgG-ELISAs, and OMP/IgG-ELISAs, respectively. Among the cut-off OD values of all the modified ELISAs, the modified ELISA tests for the WCP/IgG-ELISA, TMP/IgG-ELISA, and OMP/IgG-ELISA using the serovar Dadas isolate yielded the highest specificity, while maintaining good sensitivity in the ROC analysis.

The results of the Pearson correlation of MAT antibody titres and the IgG antibody levels under modified ELISAs in dog sera with MAT titres above a 1:20 dilution are shown in Appendix A. Overall, the antibody levels in dogs with MAT titres above a 1:20 dilution from the MAT and ELISA positively correlated with those under all modified ELISA platforms. IgG antibodies from the WCP-Mini13/IgG-ELISA, TMP-Mini06/IgG-ELISA, and OMP-Paidjan/IgG-ELISA had the most positive correlation to the MAT antibody titre among the IgG-ELISAs.

The agreement analysis of serum PCR and the MAT against all modified ELISAs is shown in Appendix A. The degree of concordance from a comparison of a PCR assay from urine and all modified ELISAs demonstrated that only the WCP-/IgG-ELISA using serovars Dadas, Bataviae, and Mini13, and the OMP-Dadas/IgG-ELISA showed slight agreement (κ = 0.10–0.20), while the other modified ELISAs represented fair agreement (κ = 0.21–0.40). On the other hand, a comparison of the serum PCR results with those of all modified ELISAs demonstrated that the TMP-Mini06/IgG-ELISA and OMP/IgG-ELISA from serovar Bataviae and Mini06 had moderate agreement (κ = 0.41–0.60), whereas the others showed fair agreement—except for the WCP-Dadas/IgG-ELISA, which only had slight agreement. In contrast, a comparison of the MAT assay with all modified ELISAs showed that the seven modified ELISAs, including the WCP/IgG-ELISA using the serovars Dadas, Mini06, and Mini13, the TMP/IgG-ELISA using the serovars Paidjan and Bataviae, and the OMP/IgG-ELISA from serovars Dadas and Paidjan showed fair agreement with the MAT, while the remaining modified ELISAs displayed slight agreement.

## 4. Discussion

Leptospirosis is a zoonotic disease that poses a threat to both human and animal health. Dogs can act as asymptomatic carriers of the pathogen, making them potential sources of infection for humans [1,32]. To effectively combat leptospirosis, it is essential to have reliable diagnostic tools and vaccines for surveillance and prevention, respectively. Serological techniques, including ELISAs, have been developed to diagnose and screen for leptospirosis in dogs [42,53,59,60,61]. This study aimed to improve the sensitivity and specificity of antibody detection in leptospirosis in our region by proposing modified indirect ELISAs incorporating local isolates of *Leptospira* serovars isolated from asymptomatic dogs in Thailand.

The leptospiral serum antibody content of animals reflects exposure to immunodominant *Leptospira* antigens during infection and vaccination. Leptospiral surface proteins are often cross-reactive, while the surface lipopolysaccharide (LPS) components are mostly serovar-specific and generate antibodies specific for those serovars [62,63,64,65,66]. Previous studies on antibody detection in dogs have used sera from animals that have either been infected or have received serovar-specific vaccines [42,53,57,65,66,67]. In the current study, the canine sera used were chosen from various types of dogs from areas where there were different circulating serovars. This allowed an analysis of serovar-specific and non-serovar-specific antibodies generated against local isolates of leptospiral serovars. The five groups of dogs represented those from an area endemic for local isolates (groups 1 and 2) and those where these isolates have not been recorded (groups 3–5). This allowed us to evaluate and confirm the modified ELISAs’ diagnostic performance using local leptospiral serovars for detecting corresponding antibodies in dogs.

Group 5, including unvaccinated puppies from non-endemic regions, was chosen as the negative control group for a serological investigation since the dogs had not been vaccinated and tested negative under PCR, confirming the absence of the pathogen in them. This group provides a clear baseline for seronegative samples, which is crucial for determining the cut-off in serological tests. The MAT was not used for these sera because the MAT is typically reserved for cases with uncertain pathogen presence or to measure antibody response, and these puppies were already confirmed as negative under PCR, a method that directly detects a pathogen’s genetic material with high sensitivity and specificity, rendering further MAT testing unnecessary [68,69]. The reliability of the PCR in confirming the absence of the pathogen justifies its use over the MAT, which, despite its high specificity, has lower sensitivity and can yield false negatives in early infection stages or in patients with low antibody levels. Therefore, the PCR-negative status of Group 5 puppies ensures a valid negative control for establishing serological test cut-offs [68,69,70,71].

Determining an optimal cut-off value for ELISA tests is essential for accurately distinguishing between positive and negative results in detecting antibodies against *Leptospira* spp. The method of setting the cut-off value at the average optical density (OD) plus four standard deviations (SD) is highlighted for its ability to improve the specificity of the test, ensuring that positive results truly reflect the presence of the disease [55,56,57,58]. This approach is supported by a study in which use of the ELISA gave high specificity (95.6%) and sensitivity (100%) compared with those of the MAT for detecting leptospirosis in dogs, by adopting the mean OD + 4SD as the cut-off value [57]. This method effectively minimises the risk of false positives, which is crucial for reliable diagnostics in a clinical setting [72].

Using local isolates of *Leptospira* serovars as protein antigens in the modified ELISAs significantly improved the accuracy of serological detection for leptospirosis in dogs in Thailand. Previous studies also have demonstrated the efficacy of ELISAs using local serovars, such as the *L. interrogans* serovar Canicola, in detecting leptospiral antibodies and increasing sensitivity and specificity compared with those of the standard MAT [42,53]. Similar results have been observed in leptospirosis studies in dogs and cattle using local serovars and isolates from endemic areas, such as *L. fainei*, serovar Hurstbridge, strain BUT 6T, and *L. interrogans*, serovar Hardjo [60,73]. These findings and the present study indicate that using known or local serovars in ELISA testing enhances the diagnostic accuracy of serological antibody detection.

The ELISAs modified using local isolates of *Leptospira* serovars demonstrated the ability to detect IgG antibodies and differentiate their levels in sera between different groups of dogs, particularly in infected dogs (group 1) that did not produce the antibody titres detected in the MAT but showed positive results via PCR on urine and serum. This is consistent with previous studies that have shown the effectiveness of modified ELISAs using local isolates of *Leptospira* serovars in detecting antibodies in negative sera and in sera shown to be positive under the MAT [42,53,54,60,73]. Unvaccinated dogs from endemic areas (Group 2) and vaccinated dogs from non-endemic areas (Group 3) exhibited different ELISA results and antibody levels compared with infected dogs (Group 1). While serum PCR results were negative for groups 2 and 3, MAT and ELISAs showed positive titres and antibody levels in some animals. The positive animals in Group 2 might have developed antibodies from previous exposure to local leptospiral isolates in the endemic area. In contrast, antibodies in Group 3 could have been generated via exposure to serogroup cross-reactive antigen proteins from vaccine serovars. In diagnosing leptospirosis, it is well recognised that there is no direct correlation between the results of tests that detect the agent, such as PCR, and serological testing that provides evidence of past exposure [38,44]. However, combining PCR results with serological test results can improve the effectiveness of leptospirosis diagnosis [74,75,76,77]. Using ELISAs with local serovars and isolates may enhance the diagnostic efficacy of serological tests, especially in samples that are PCR-negative and negative for conventional antibody detection under the MAT.

The modified ELISAs employed different protein components as antigens, including WCP, TMP, and OMP preparations. WCPs of leptospiral cells contain many proteins derived from the whole and outer membranes. On the other hand, TMPs consist of cytoplasmic membrane and outer membrane proteins. The heat shock proteins GroEL and DnaK, which are primarily present in cytoplasmic membranes, are potential immunoreactive protein antigens in these preparations. The OMPs, which include numerous possible highly immunogenic proteins, such as LipL32, LipL45/31, and LipL41, are the most conserved components across all leptospiral serovars and species. In contrast, lipopolysaccharides have a wide range of carbohydrate side chains, influencing antigenic diversity across several leptospiral serovars [47,48,49,50,51,52]. Among these, the ELISAs based on WCPs exhibited higher sensitivity, followed by those based on TMPs and OMPs. The WCP-based ELISAs are better suited for antibody surveillance due to their high sensitivity; nevertheless, this enhanced sensitivity can frequently result in false-positive results. Although ELISAs based on total membrane proteins (TMPs) and outer membrane proteins (OMPs), especially OMP-based ELISAs, demonstrated a high level of specificity and are therefore suitable for antibody screening and diagnosis to differentiate between infection and no infection or vaccination in dogs, they tend to yield a high percentage of false-negative results. False-positive and false-negative serological reactivities reflect the sensitivity and specificity of each method via the synthesis of protein antigens and serovars used in the ELISAs. These findings can be attributed to the differences in protein components and the impact of various serovars on the sensitivity and specificity of the ELISAs [42,48,53,61].

Agreement analysis between the ELISAs modified using local isolates of *Leptospira* serovars and the PCR test in this study indicated a slight to moderate degree of concordance, suggesting a reliable correlation between the two methods. The agreement between the ELISAs and MAT ranged from slight to fair, highlighting more satisfactory agreement between PCR-based antigen detection and ELISA-based antibody detection using local serovars and isolates compared with the MAT using reference serovars combined with local serovars and isolates [73,78,79]. To enhance the accuracy of serological diagnostic testing for antibody detection and monitoring in canine leptospirosis, the utilisation of local serovars or isolates in combination with appropriate antigen preparations is crucial. Thus, the identification of the local serovars and isolates in each area, as well as the serovars identified via serotyping, is essential and should be explored.

The discrepancies observed between the ELISA and the MAT results in diagnosing leptospirosis are primarily due to the different antigenic profiles that each test targets and the stages of disease at which they are most effective. The ELISA is designed to detect antibodies against specific antigens or epitopes, which may be highly specific to certain strains of a pathogen, while the MAT detects antibodies against a broader range of antigens presented by live bacteria, including surface proteins not targeted by the ELISA [80]. This can lead to discrepancies, particularly if the ELISA and MAT do not include antigens and leptospires from the strains circulating in each region, potentially resulting in false negatives. The World Health Organization recommends using a locally optimised MAT panel that represents the currently circulating strains to improve sensitivity [81]. The discrepancies between these tests underscore the importance of the regional customisation of diagnostic tests to enhance accuracy, which can be achieved by including antigens from local strains in ELISAs or by optimizing MAT panels with these strains [80,81].

The research highlights the importance of finding and using local strains of *Leptospira* from dogs to improve diagnostic tests in regions with high prevalence like Thailand where leptospirosis presents significant public health and economic difficulties. The CDC stresses the significance of national surveillance for controlling leptospirosis, a condition that must be reported nationally in several countries. This study proposes that creating ELISA plates specific to the most common local serovars could enhance diagnostic precision, given the current diagnostic methods’ shortcomings in accurately detecting the disease due to their low specificity and the risk of cross-reactivity with other tropical diseases. A detailed plan for consistent monitoring and readiness for diagnosis, which involves creating a collection of serovars, regularly assessing the composition of ELISA plates, improving surveillance programs, developing ELISAs based on recombinant antigens, and fostering collaboration and data exchange, is suggested to direct future diagnostic methods and improve public health outcomes in the area [32,44,82].

The introduction of new serovars in tests for leptospirosis has the potential to greatly improve the accuracy of diagnostic tests, which are now limited by the specificity and sensitivity of current assays. Integrating the new serovars with the authorised set may enhance diagnostic accuracy and epidemiological surveillance, perhaps establishing them as the standard for leptospirosis diagnosis and surveillance, pending additional research and validation studies. This method would require revisions to current regulations and guidelines to include the new genotypes and serovars. If the new set is not officially adopted, it can still be used in conjunction with the approved set to improve surveillance and research, helping to understand the genetic diversity and evolution of *Leptospira* spp. Adopting new serovars requires regulatory compliance, integration with current processes, improved epidemiological data gathering, heightened public health awareness, and perhaps revised immunisation approaches. Both sets of serovars can be used together, with the new serovars being utilised for more accurate and quicker diagnosis. This development offers a chance to enhance the comprehension and control of leptospirosis by improving diagnostic precision and epidemiological monitoring [83,84,85,86,87,88].

Overall, the findings of this study contribute to the advancement of leptospirosis surveillance and emphasise the importance of using local isolates of *Leptospira* serovars in the design of effective serological techniques. Further research and validation studies are warranted to confirm the efficacy and applicability of these indirect ELISAs modified using local *Leptospira* serovar isolates in diverse geographical locations and dog populations.

## 5. Conclusions

In conclusion, this study demonstrates the effectiveness of ELISAs modified using local isolates of leptospiral serovars in improving the sensitivity and specificity of antibody detection for leptospirosis in dogs. Using protein antigens extracted from local isolates of leptospiral serovars enhanced the diagnostic accuracy of the serological tests, particularly in samples that were PCR-negative and negative for conventional antibody detection under the MAT. The choice of protein antigen preparation, such as WCPs, TMPs, or OMPs, affects the sensitivity and specificity of the ELISAs. The WCP-based ELISAs offer higher sensitivity, while the OMP-based ELISAs provide higher specificity. The ELISA modified with the combination of local isolates of leptospiral serovars with appropriate antigen preparations can improve the accuracy of serological diagnostic testing for antibody detection and monitoring in canine leptospirosis. The importance of the identification of local circulating leptospiral serovars in certain regions needs to be determined, and has practical implications for achieving more accurate diagnosis and management of the disease in endemic areas to prevent leptospirosis in animals and humans.

## Figures and Tables

**Figure 1 animals-14-00893-f001:**
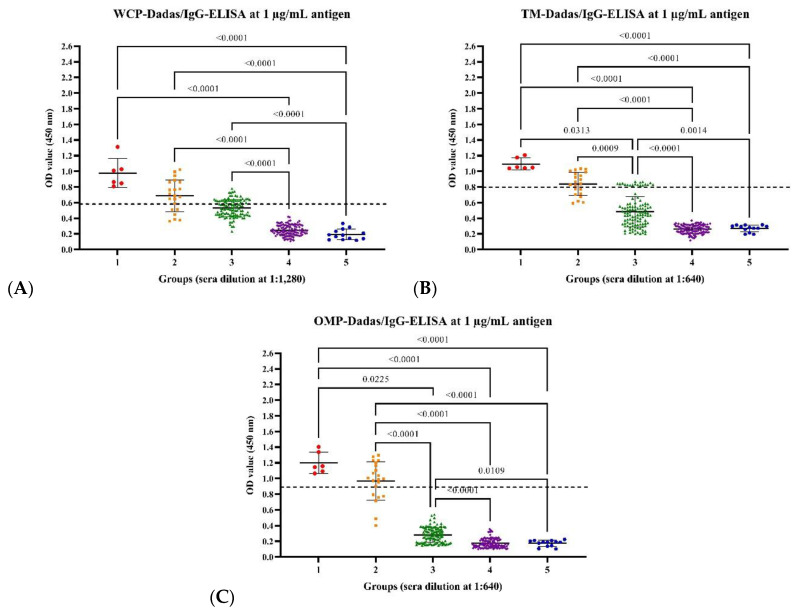
The levels of IgG antibody detected in modified ELISAs against whole-cell protein (WCP), total membrane protein (TMP), and outer membrane protein (OMP) from the local Thai isolate of *Leptospira* serovar, including serovar Dadas, at 1:1280 (**A**), 1:640 (**B**), and 1:640 (**C**) sera dilutions. Comparisons among 260 sera from five groups consisting of dogs from Nan Province confirmed as infected via PCR and isolation (Group 1), unvaccinated dogs from Nan Province (Group 2), vaccinated dogs from Bangkok (Group 3), unvaccinated dogs from non-endemic areas (Group 4), and unvaccinated puppies from non-endemic areas (Group 5). The significant differences of the IgG antibody levels between dog sera group were analyzed using Kruskal–Wallis test and Dunn’s post hoc test (*p*-value < 0.05). (**A**–**C**) demonstrated the modified ELISAs against WCP, TMP and OMP from serovar Dadas identified that dogs from endemic area (groups 1 and 2) have the higher level of IgG antibody than dogs in non-endemic areas (groups 3, 4, and 5), with a highly significant difference.

**Figure 2 animals-14-00893-f002:**
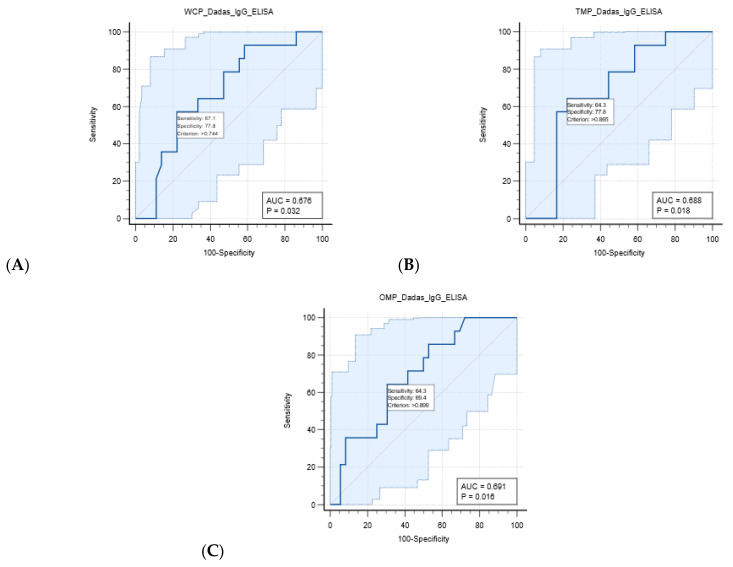
Receiver–operator curve (ROC) and area under the curve of ROC (AUC) of modified ELISAs against whole-cell protein (WCP), total membrane protein (TMP), and outer membrane protein (OMP) from the local Thai isolates of *Leptospira* serovars, including serovar Dadas, at 1:1280 (**A**), 1:640 (**B**), and 1:640 (**C**) sera dilutions, with 50 sera from three groups consisting of dogs from Nan Province confirmed as infected via PCR and isolation (Group 1), unvaccinated dogs from Nan Province (Group 2), and vaccinated dogs from Bangkok (Group 3). All the ROC curve and AUC of the ROC of modified ELISAs were analyzed by MedCalc software. (**A**–**C**) displayed the sensitivity and specificity of the modified ELISAs against WCP, TMP and OMP from serovar Dadas using the cut-off OD values that were set by ROC analysis showed the moderate sensitivity, specificity and AUC, with *p*-value < 0.05.

**Table 1 animals-14-00893-t001:** Names of 29 leptospiral isolates, including 24 representative reference isolates and 5 local Thai isolates, that were used in the microscopic agglutination test (MAT) and enzyme-linked immunosorbent assay (ELISA) in the study. These isolates were associated with 27 serovars (serovar Bataviae and Mini each had two different isolates).

Serovars/Isolates	MAT	ELISA
24 representative reference isolates (belonging to 24 serovars)		
Australis	Used	-
Aumtumnalis	Used	-
Ballum	Used	-
Bataviae	Used	-
Canicola	Used	Used
Cellidoni	Used	-
Cynopteri	Used	-
Djasiman	Used	-
Grippotyphosa	Used	Used
Hebdonadis	Used	-
Icterohaemorrhagiae	Used	Used
Javanica	Used	-
Louisaina	Used	-
Manhao	Used	-
Mini	Used	-
Panama	Used	-
Pomona	Used	Used
Pyrogenes	Used	-
Ranarum	Used	-
Sarmin	Used	-
Sejroe	Used	-
Shermani	Used	-
Tarasovi	Used	-
Semaranga	Used	-
5 local Thai isolates (belonging to 4 serovars)		
Paidjan strain CUDO5	Used	Used
Dadas strain CUDO8	Used	Used
Bataviae strain D64	Used	Used
Mini strain CUDO6	Used	Used
Mini strain CUD13	Used	Used

**Table 2 animals-14-00893-t002:** Information about the five dog groups, and the number and percentage of 260 sera from the groups that tested positive via urine isolation (agent detection), urine and serum PCR (agent detection), the MAT (antibody detection), and modified ELISAs (antibody detection) using a local leptospiral serovar isolate.

Groups	Group 1	Group 2	Group 3		Group 4	Group 5	Total
	Infected Dogs from Nan Province That Were Confirmed as Infected via Positive PCR and Isolation	Unvaccinated Dogs from Nan Province	Vaccinated Dogs from Non-Endemic Areas		Unvaccinated Dogs from Non-Endemic Areas	Unvaccinated Puppies from Non-Endemic Areas	
Number of samples	6	21	23 *	89 **	108	13	260
			112				
Area	Endemic area: Nan Province	Endemic area: Nan Province	Non-endemic area: Bangkok, Samut Prakan, and Chonburi provinces		Non-endemic area: Bangkok, Samut Prakan, and Chonburi provinces	Non-endemic area: Bangkok, Samut Prakan, and Chonburi provinces	-
Age	>1 year	>1 year	>1 year		>1 year	Two months	-
Vaccination status	No vaccination	No vaccination	Complete vaccination		No vaccination	No vaccination	-
Methods (Nested PCR, Isolation, and MAT assays)							
Nested PCR from urine	6 (6/6; 100%)	0 (0/21; 0%)	N/A		N/A	N/A	6 (6/27; 22%)
Isolation from urine	4 (4/6; 67%)	0 (0/21; 0%)	N/A		N/A	N/A	4 (4/27; 15%)
Nested PCR from sera	6 (6/6; 100%)	0 (0/21; 0%)	0 (0/112; 0%)		0 (0/108; 0%)	0 (0/13; 0%)	6 (6/260; 2%)
MAT from sera	0 (0/6; 0%)	0 (0/21; 0%)	0 (0/23; 0%)	N/A	N/A	N/A	14 (14/50; 28%)
Methods (modified ELISAs), protein preparation, and local isolates of serovars used in the ELISAs							
WCP-Dadas/IgG-ELISA	6 (6/6; 100%)	14 (14/21; 67%)	12 (12/23; 52%)	25 (25/89; 28%)	0 (0/108; 0%)	0 (0/13; 0%)	57 (57/260; 22%)
			37 (37/112; 33%)				
TMP-Dadas/IgG-ELISA	6 (6/6; 100%)	13 (13/21; 62%)	6 (6/23; 26%)	12 (12/89; 13%)	0 (0/108; 0%)	0 (0/13; 0%)	37 (37/260; 14%)
			18 (18/112; 16%)				
OMP-Dadas/IgG-ELISA	6 (6/6; 100%)	15 (10/21; 48%)	0 (0/23; 0%)	0 (0/89; 0%)	0 (0/108; 0%)	0 (0/13; 0%)	21 (21/260; 8%)
			0 (0/112; 0%)				

N/A: Not applicable. WCP: whole-cell protein; TMP: total membrane protein; OMP: outer membrane protein; *: Twenty-three serum samples from group 3 were examined via both the MAT and ELISAs, and used for the preliminary evaluation of the effectiveness of diagnostic performance of the modified ELISAs. **: Another 89 serum samples were examined only via ELISAs and were used to screen for leptospiral IgG antibody and confirm the usefulness of the modified ELISAs.

**Table 3 animals-14-00893-t003:** Frequency distribution of serum antibody titres in the microscopic agglutination test (MAT) with 29 different isolates of *Leptospira* using 50 sera from three groups of dogs (groups 1, 2, and 3).

Groups	Number	MAT	Serovars and Serological Titres in MAT				
			Hebdonadis	Sejroe	Shermani		Paidjan (CUDO5)
Isolation and PCR-confirmed infected dogs from Nan Province (Group 1)	6	0 (0/6; 0%)	-	-	-		-
Unvaccinated dogs from Nan Province (Group 2)	21	11 (11/21; 52%)	1 *	1	9 *		1
			1:20	1:80	4	5	
					1:20	1:40	
Vaccinated dogs from Bangkok (Group 3)	23	3 (3/23; 13%)	-	-	3		-
					1:20		
Total	50	14 (28%)	1	1	12		1

* One sample from an unvaccinated dog from Nan Province (Group 2) had an MAT titre more than or equal to 1:20 for the two leptospiral serovars Hebdonadis and Shermani. The MAT was performed using 29 different isolates of *Leptospira* consisting of 24 representative reference isolates and 5 local Thai isolates. These isolates belonged to 27 serovars, of which 2 serovars, Bataviae and Mini, each included 2 different isolates.

## Data Availability

The data presented in this study are available on request from the corresponding author. The data are not publicly available due to [some data containing information that could compromise the privacy of research participants, including another ongoing study].

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
