# Peer review of "Improved Antibody Detection for Canine Leptospirosis: ELISAs Modified Using Local Leptospiral Serovar Isolates from Asymptomatic Dogs"

_animals, 2024, doi:10.3390/ani14060893_

Round 1

Reviewer 1 Report

Comments and Suggestions for Authors

The objective of this study was the investigation of the use of local Leptospira strains of different serovars from dogs, in a modified indirect ELISA can improve the accuracy of detection of the infection when compared to the approved standard MAT test.

There is some merit in the manuscript, and I support its publication, but the authors should undertake extensive changes before resubmission.

-The manuscript is extremely lengthy and really tiresome for readers. The authors should reduce significantly its size to at least half of the current extent, ideally even more. The provision of full details of the results should be commended, because it contributes to clarity and tranparencey, but this will be done rarely by future readers. Hence, authors should move large parts of the findings in supplementary material. This will not reduce the interest of the manuscript, but it will make it more attractive to readers. For example, Table 4 and Figure 1 and 2 only take space and do not help readers. Please move to supplementary material. Also, the introduction is extremely lengthy and should be reduced in size.

-Please do not talk about strains but about isolates.

-There is an issue that the authors do not address adequately. The authors indicate that their new set is better than the approved set. How do they propose to go forward from this point onwards? Should a new set of serotypes be approved and new regulations be produced? If no new set is approved, how they move forward? Will they use both sets for an examination? Will they use only the new set and ignore the approved one? This is an important topic and the authors must address it in the discussion.

-Also, please add one sentence in the conclusions about the clinical significance of the findings.

Overall. Interesting manuscript, but needs significant reduction in size before acceptance.

Author Response

Dear Editor of Animals, the MDPI journal

              Thank you very much for suggesting our manuscript "Improved the susceptibility of antibody detection for canine leptospirosis: modified ELISAs using local strains of leptospiral serovars isolated from asymptomatic dogs." We are very impressed by your kind response and the sophisticated system. All suggestions have been carefully responded to topic by topic and noted in blue letters relevant to the revised manuscript. Hopefully, our revision will be satisfactory enough to be published in Animals, the MDPI journal.

                                                                                                                Sincerely yours,

                                                                                                                 Prapasarakul N.

Reviewer 1

Comments and Suggestions for Authors

The objective of this study was the investigation of the use of local Leptospira strains of different serovars from dogs, in a modified indirect ELISA can improve the accuracy of detection of the infection when compared to the approved standard MAT test. There is some merit in the manuscript, and I support its publication, but the authors should undertake extensive changes before resubmission.

  1. The manuscript is extremely lengthy and really tiresome for readers. The authors should reduce significantly its size to at least half of the current extent, ideally even more. The provision of full details of the results should be commended, because it contributes to clarity and transparency, but this will be done rarely by future readers. Hence, authors should move large parts of the findings in supplementary material. This will not reduce the interest of the manuscript, but it will make it more attractive to readers. For example, Table 4 and Figure 1 and 2 only take space and do not help readers. Please move to supplementary material. Also, the introduction is extremely lengthy and should be reduced in size.

Thank you for your valuable comments.

According to the aims of the study that was developed the modified indirect ELISAs by utilizing the new set of the local leptospiral isolates of the different serovars from an asymptomatic dog in Thailand, with the differences of protein antigen preparations, and investigated how the modified ELISAs  detect and differentiate the infection status and surveillance of the leptospiral antibody among Thai dog groups, and improve the diagnostic performance, including sensitivity, specificity, and accuracy, compared to several methods, consisting of urine isolation (agent detection), urine and serum PCR (agent detection), and approved standard MAT (antibody detection).

The manuscript emphasized to show the results how the local isolates of the leptospiral serovars can detect the antibody in the serum samples, especially infected dogs, that MAT cannot detect, and differentiate the antibody status among dog group between endemic area and non-endemic areas of the local isolates of leptospiral serovars. Also, the manuscript highlighted the discriminatory power and diagnostic performance of the modified ELISAs with the different protein antigen preparations for selection to practical use in further investigation and proved that side by side with the use of the vaccine serovars in other modified ELISAs.

The revised manuscript was changed following the reviewer suggestions, as size reduction of the introduction part and move some of the Table and Figure results to supplementary material. The introduction part of the revised manuscript was reduced its size by cutting the small, detailed data off and keeping the important background story and information. (Page 2 and 3, Part 1: Introduction, Line 69-133). For the unattractive results, such as Figure 4, and Figure 1 and 2, were removed to the supplementary materials, as the supplementary table 1, and supplementary figures 2 and 3. (Page 27-30 and 33-36, Part 6: Supplementary materials, Line 683-688 and 712-738). However, there are a little bit changes about Table 2 and Figure 1 that the authors want to keep some results about the modified ELISAs with the selected local serovar, as serovar Dadas (serogroup Grippotyphosa) that excluded from the standard MAT, by including and approving in Table 2 besides the other leptospiral diagnostic tests and show the discriminatory power of the modified ELISAs with the different antigen preparations among all dog groups from endemic area and non-endemic areas in Figure 1. (Page 10-11 and 20, Part 3: Results, Line 355-361 and 442-450).

  1. Please do not talk about strains but about isolates.

Thank you for your valuable comment. The word “strains” was replaced with “isolates”, as the reviewer suggested throughout the entire revised manuscript.

  1. There is an issue that the authors do not address adequately. The authors indicate that their new set is better than the approved set. How do they propose to go forward from this point onwards? Should a new set of serotypes be approved, and new regulations be produced? If no new set is approved, how do they move forward? Will they use both sets for an examination? Will they use only the new set and ignore the approved one? This is an important topic and the authors must address it in the discussion.

            Thank you for your valuable comments. The discussion of this topic was updated and added to the revised manuscript in the discussion part. (Page 25, Part 4: Discussion, Line 646-660).

            The implications of the new serotype set, the introduction of a new set of serotypes for leptospirosis could enhance the precision of diagnostic tests, which are currently limited by the specificity and sensitivity of existing assays. This advancement could lead to more accurate diagnosis and timely clinical interventions. Furthermore, the new serotypes could provide insights into the relative importance of different reservoir species in the transmission of human and animal leptospirosis, essential for developing effective control strategies.

  • Moving forward with new serotypes
  • Integration of both sets for examination. The authors suggest combining the new set with the existing approved set to improve the accuracy of diagnosis and epidemiological monitoring of leptospirosis. This method would not eliminate the authorized collection but enhance it by incorporating new discoveries to encompass a wider range of genetic types and serotypes.
  • Adoption of the new set with regulatory updates. If further research and validation studies confirm the superiority of the new set, it may result in the new set being adopted as the standard for diagnosing and monitoring leptospirosis. Updating current legislation and standards is required to include novel genotypes and serotypes, along with potentially new diagnostic methods utilizing MLST and serovar identification.
  • Continued Research and Validation. Additional study is required to confirm the clinical significance of the new genotypes and serotypes and their impact on disease control and prevention methods before implementing any changes. Comparative studies could be conducted comparing the new and authorized sets in different endemic regions to evaluate their effectiveness in recognizing and monitoring disease outbreaks.
  • If no new set is approved

If the new set is not officially adopted, researchers can still use it alongside the approved set for improved surveillance and study. The new collection can be used as a supplementary tool to comprehend the genetic variety and evolution of Leptospira spp., assisting in the creation of more efficient preventative and control strategies. The information obtained from using both sets can help in updating the approved set to ensure its relevance and effectiveness in identifying the microorganisms causing leptospirosis outbreaks.

  • Potential Pathways Forward.

The adoption of new serotypes requires careful consideration of several factors:

  • Regulatory Considerations: Regulatory agencies need to assess and authorize new diagnostic tests that are developed for the new serotypes. This method involves demonstrating the precision, reliability, and clinical significance of the assays.
  • Integration with Existing Practices: New serotypes need to be included into existing diagnostic techniques like the Microscopic Agglutination Test (MAT) and IgM-based ELISAs, which are often utilized despite their limitations. Integration may include cross-validation studies to guarantee consistency and comparability of outcomes.
  • Epidemiological Data Collection: Implement advanced surveillance techniques to gather epidemiological data utilizing the new serotypes. This data will be essential for monitoring the disease's spread and identifying hotspots for targeted responses.
  • Public Health Awareness: Enhanced public health awareness and education on the new serotypes and their implications are essential for healthcare providers and the public to comprehend the advantages of the new diagnostic tools.
  • Vaccination Strategies: It may be necessary to update or create vaccinations to incorporate novel serotypes, especially if they are strains not already covered by existing vaccines. Vaccination methods should consider the frequency and spread of the new serotypes.
  • Using Both Sets of Serotypes in Practice

            Both the new and old sets of serotypes could be used together in a complementary way. The new serotypes can be used for accurate and quick diagnosis, but the current serotypes can still be utilized for wider epidemiological monitoring and vaccine development until the new serotypes are completely incorporated into those areas.

            In conclusion, the identification of new genetic variants of Leptospira spp. offers a chance to improve the comprehension and control of leptospirosis. By combining both sets for analysis, updating the new set with regulatory changes, or doing more study and validation, these discoveries could enhance the accuracy of diagnosis and epidemiological monitoring of this zoonotic illness.

  1. Also, please add one sentence in the conclusions about the clinical significance of the findings.

Thank you for your valuable comment. The discussion of this topic was updated and added to the revised manuscript in the discussion part. (Page 25, Part 5: Conclusions, Line 667-681).

  1. Interesting manuscript but needs significant reduction in size before acceptance.

Thank you for your valuable comment. The revised manuscript was reduced the size of the results showed by focusing on the objectives of the study and the results that proved the hypothesis and took some advantages to better understand for the readers, as well as reminded the important of the isolates of local circulating leptospiral serovar. The other information and results that slightly influenced the objectives of the study were moved to the supplementary material part.

References

Bertasio, C., Boniotti, M. B., Lucchese, L., Ceglie, L., Bellinati, L., Mazzucato, M., . . . Natale, A. (2020). Detection of New Leptospira Genotypes Infecting Symptomatic Dogs: Is a New Vaccine Formulation Needed? Pathogens, 9(6). https://doi.org/10.3390/pathogens9060484

Matthias, M. A., Lubar, A. A., Lanka Acharige, S. S., Chaiboonma, K. L., Pilau, N. N., Marroquin, A. S., . . . Vinetz, J. M. (2022). Culture-Independent Detection and Identification of Leptospira Serovars. Microbiol Spectr, 10(6), e0247522. https://doi.org/10.1128/spectrum.02475-22

Mgode, G. F., Machang’u, R. S., Mhamphi, G. G., Katakweba, A., Mulungu, L. S., Durnez, L., . . . Belmain, S. R. (2015). Leptospira Serovars for Diagnosis of Leptospirosis in Humans and Animals in Africa: Common Leptospira Isolates and Reservoir Hosts. PLOS Neglected Tropical Diseases, 9(12), e0004251. https://doi.org/10.1371/journal.pntd.0004251

Miotto, B. A., Guilloux, A. G. A., Tozzi, B. F., Moreno, L. Z., da Hora, A. S., Dias, R. A., . . . Hagiwara, M. K. (2018). Prospective study of canine leptospirosis in shelter and stray dog populations: Identification of chronic carriers and different Leptospira species infecting dogs. PLoS One, 13(7), e0200384. https://doi.org/10.1371/journal.pone.0200384

Samrot, A. V., Sean, T. C., Bhavya, K. S., Sahithya, C. S., Chan-Drasekaran, S., Palanisamy, R., . . . Mok, P. L. (2021). Leptospiral Infection, Pathogenesis and Its Diagnosis-A Review. Pathogens, 10(2). https://doi.org/10.3390/pathogens10020145

Smith, A. M., Stull, J. W., & Moore, G. E. (2022). Potential Drivers for the Re-Emergence of Canine Leptospirosis in the United States and Canada. Trop Med Infect Dis, 7(11). https://doi.org/10.3390/tropicalmed7110377

Zhang, C., Yang, H., Li, X., Cao, Z., Zhou, H., Zeng, L., . . . Jiang, X. (2015). Molecular Typing of Pathogenic Leptospira Serogroup Icterohaemorrhagiae Strains Circulating in China during the Past 50 Years. PLOS Neglected Tropical Diseases, 9(5), e0003762. https://doi.org/10.1371/journal.pntd.0003762

Reviewer 2 Report

Comments and Suggestions for Authors

Thank you for this huge work in Leptospirosis ELISA development.

I have a couple of questions.

1) Why do you use group 5 as the negative group (Unvaccinated and PCR negative) for the determination of the cut-off? These sera were not tested using MAT but MAT is supposed to be the reference method.

2) You determined the cut-off value as the average OD + 4SD. Please cite a reference of the literature for this point.  Why not make a ROC curve? Why not determine a gray zone?

3) The agreement between ELISA and PCR is low but it is quite normal because these methods don't detect the same analytes. However, the mis-agreement between ELISA and MAT is a little bit more surprising. Is it due to the type of exposed antigens in both methods?

4) In your strategy, it is important to be able to determine the "local" serovars. Is there any national surveillance for this?  Do you recommend preparing ELISA plates for the most common serovars in Thailand?  Is it feasible in practice? I suggest you propose a concrete strategy at the end of the paper (Discussion or conclusion).

Author Response

Dear Editor of Animals, the MDPI journal

              Thank you very much for suggesting our manuscript "Improved the susceptibility of antibody detection for canine leptospirosis: modified ELISAs using local strains of leptospiral serovars isolated from asymptomatic dogs." We are very impressed by your kind response and the sophisticated system. All suggestions have been carefully responded to topic by topic and noted in blue letters relevant to the revised manuscript. Hopefully, our revision will be satisfactory to be published in Animals, the MDPI journal.

                                                                                                                Sincerely yours,

                                                                                                                 Prapasarakul N.

Reviewer 2

Comments and Suggestions for Authors

Thank you for this huge work in Leptospirosis ELISA development.

I have a couple of questions.

  1. Why do you use group 5 as the negative group (Unvaccinated and PCR negative) for the determination of the cut-off? These sera were not tested using MAT but MAT is supposed to be the reference method.

Thank you for your valuable comment. The discussion of this topic was updated and added to the revised manuscript in the discussion part. (Page 23, Part 5: Discussion, Line 534-546).

            Group 5 (Unvaccinated puppy dogs from non-endemic areas) was chosen as the negative control group because it consists of the history of no vaccination and PCR-negative, which are presumed to be free from the pathogen in question. This group provides a clear baseline for seronegative samples, which is essential when determining the cut-off for a serological test. The rationale behind not using the Microscopic Agglutination Test (MAT) for these sera is that MAT is typically employed as a reference method for antibody detection in cases where the presence of the pathogen is uncertain or where antibody response is being measured (Chirathaworn et al., 2014). In Group 5, the puppies were verified to be PCR-negative, indicating the absence of the pathogen's genetic material. PCR's high sensitivity and specificity can conclusively verify the pathogen's absence, eliminating the need for additional testing with MAT for these specific samples (Martin et al., 2022). Moreover, the unvaccinated status of these puppies makes them suitable as negative controls since they have not been exposed to the antigen in the vaccination that could trigger an antibody response detectable by serological testing.

         PCR is a reliable test to confirm the absence of the pathogen, as shown by other studies that detect leptospires in blood and urine samples. A negative result indicates the absence of leptospires. Although MAT exhibits good diagnostic specificity, its sensitivity is very modest, leading to potential false negatives, particularly during the initial infection phases or when antibody levels are low. (Chirathaworn et al., 2014; Martin et al., 2022). Therefore, using PCR-negative samples as a negative control is an acceptable method for determining a threshold for serological testing, as it confirms that the baseline samples are genuinely free of the pathogen. Group 5, consisting of unvaccinated puppies from non-endemic areas, is a suitable negative control because to their unvaccinated status and validated PCR-negative results. This establishes a dependable baseline for seronegative status without requiring MAT confirmation.

  1. You determined the cut-off value as the average OD + 4SD. Please cite a reference of the literature for this point. Why not make a ROC curve? Why not determine a gray zone?

Thank you for your valuable comments. The citation for using the average OD + 4SD method in the materials and methods part of the revised manuscript. (Page 7, Part 2: Materials and Methods, Line 283-307). The discussion of this topic was updated and added to the revised manuscript in the discussion part. (Page 23, Part 5: Discussion, Line 547-556). The ROC analysis was added to the materials and methods part of the revised manuscript to enhance the information about the diagnostic performances of the modified tests. (Page 21 and 36-38, Part 3 and 6: Results and Supplementary materials, Line 451-458 and 739-754).

Establishing a cut-off value for ELISA tests is essential for differentiating between positive and negative results for the existence of antibodies against Leptospira spp. The cut-off value, calculated as the average optical density (OD) plus 4 standard deviations (SD), is a technique employed to improve the specificity of the test, guaranteeing that positive results accurately signify the existence of the condition. (Gómez-Morales et al., 2016; Gottschalk et al., 1994; Ribotta et al., 2000; Woo et al., 2001). The average OD + 4SD is a validated cut-off value for determining cut-off values in ELISA testing for leptospirosis due to its specificity in similar diagnostic procedures. The investigation on detecting leptospiral antibodies in dogs used a cut-off value of the mean + 4 standard deviations to differentiate between positive and negative samples. (Ribotta et al., 2000). This method ensures a high specificity, as demonstrated in the study, where the ELISA's specificity relative to the microscopic agglutination test (MAT) was shown to be 95.6%, with a sensitivity of 100% (Ribotta et al., 2000). This approach minimizes the likelihood of false positives, which is crucial in a diagnostic setting.

The decision to not use a Receiver Operating Characteristic (ROC) curve or determine a gray zone in this study may be due to the group 5 (Unvaccinated puppy dogs from non-endemic areas) not undergoing MAT to establish a seronegative baseline for the ROC analysis. ROC curve analysis evaluates the diagnostic accuracy of a test at different threshold levels, offering information on sensitivity and specificity. The dataset and study design may not have been suitable for ROC analysis. If the dataset does not include a wide variety of borderline cases or if the study design aims to maximize specificity to prevent false positives in a low-prevalence environment, the average OD + 4SD technique may be the preferred choice. (Desakorn et al., 2012; Ribotta et al., 2000).

However, integrating ROC curve analysis could improve the diagnostic performance of the modified ELISA tests in this study by providing a more comprehensive assessment of the test's diagnostic performance across various cut-off values. This research may reveal an appropriate balance of sensitivity and specificity, potentially enhancing the test's overall diagnostic accuracy. (Desakorn et al., 2012). Determining a gray zone, or an indeterminate range, could be considered if there are a large number of samples with OD values close to the cut-off and the distinction between positive and negative is unclear, a gray zone, or ambiguous range. The lack of a gray zone in the study was owing to a clear distinction between positive and negative samples at the chosen cut-off value, which reduced the need for retesting or additional confirmatory tests. However, reconsidering the decision to include a gray zone could be useful, especially if it helps to better handle samples that lie near the cut-off, thus minimizing the chance of misclassification.(Bajani et al., 2003).

In conclusion, the choice of cut-off value and decision not to employ ROC curve analysis or identify a gray zone were most likely influenced by the study's specific aims and constraints, such as the need for high specificity and the dataset's features. Future research could use these findings to potentially improve the diagnostic value of the ELISA test for leptospirosis serodiagnosis.

  1. The agreement between ELISA and PCR is low but it is quite normal because these methods don't detect the same analytes. However, the mis-agreement between ELISA and MAT is a little bit more surprising. Is it due to the type of exposed antigens in both methods?

            Thank you for your valuable comments. The discussion of this topic was updated and added to the revised manuscript in the discussion part. (Page 24, Part 5: Discussion, Line 619-632).

            The differences observed between the Enzyme-Linked Immunosorbent Assay (ELISA) and the Microscopic Agglutination Test (MAT) in diagnosing leptospirosis can be attributed to a number of factors, the most important of which are the different antigenic profiles used in both tests and the stage of the disease at which they are most effective. The various antigenic profiles used by ELISA and MAT are one of the primary causes for the difference between their results. ELISA is often used to identify antibodies to certain antigens or epitopes, which can be extremely specific to certain disease strains. On the other hand, MAT identifies antibodies against live bacteria, which may express a greater spectrum of antigens, including surface proteins that the ELISA does not target. This discrepancy in antigenic detection might cause inconsistencies in test findings, especially if circulating strains in a region express antigen that are not included in the ELISA utilized. (Jayasundara et al., 2021). For regional specificity and strain variation, The World Health Organization suggests adopting a locally customized MAT panel that represents the currently circulating strains in a specific region to improve the sensitivity of the test. (Jayasundara et al., 2021). This recommendation emphasizes the value of geographical specificity in diagnostic tests. Pathogen strains can vary greatly between geographic locations, and diagnostic procedures such as ELISA may be designed based on strains that are common in one place but not in another. If the ELISA is based on antigens from strains that do not circulate in the test region, it may fail to detect antibodies to local strains, resulting in false-negative results. MAT, when tailored with local strains, may identify antibodies that the usual ELISA misses, hence explaining the observed difference (Jayasundara et al., 2021). For implications for diagnostic testing, the discrepancies between ELISA and MAT underscore the necessity for regional customization of diagnostic procedures. Diagnostic accuracy can be considerably increased by utilizing ELISA with antigens from local strains or by customizing the MAT panel to include these strains. This method ensures that the tests are sensitive to the specific antigenic profiles of the pathogens prevalent in the location, thus lowering the possibility of false negatives or positives (Jayasundara et al., 2021).

            In conclusion, the disagreement between ELISA and MAT results can be attributed to the different antigenic profiles targeted by these tests and the variation in pathogen strains across different regions. To mitigate these discrepancies, it is crucial to adapt diagnostic tests to the local epidemiological context, either by including region-specific antigens in ELISA or by optimizing MAT panels with locally circulating strains. This approach not only enhances the accuracy of diagnostic tests but also supports more effective disease surveillance and control efforts.

  1. In your strategy, it is important to be able to determine the "local" serovars. Is there any national surveillance for this? Do you recommend preparing ELISA plates for the most common serovars in Thailand?  Is it feasible in practice? I suggest you propose a concrete strategy at the end of the paper (Discussion or conclusion).

            Thank you for your valuable comments. The discussion of this topic was updated and added to the revised manuscript in the discussion part. (Page 24-25, Part 5: Discussion, Line 633-645).

            The finding of this study has highlight and recommendation about the use of the identified local circulating serovars from dogs, which is important to determine and used in diagnostic tests on endemic regions.

            Leptospirosis is a zoonotic illness caused by bacteria from the genus Leptospira, which consists of more than 250 harmful serovars. These serovars have unique antigenic characteristics and can differ greatly by geographical area, impacting both people and animals. This disease is a significant concern in tropical and subtropical regions, such as Thailand, because to its impact on public health and economy. National surveillance is essential for discovering and tracking the dissemination of leptospirosis serovars at a local level. The Centers for Disease Control (CDC) states that leptospirosis is a condition that must be reported nationwide in other nations, underscoring the significance of surveillance in controlling the disease. Thailand faces a substantial challenge with leptospirosis, demonstrated by the varying incidence rates and the disease's prevalence in domestic animals like livestock, highlighting the need for a strong surveillance system. (Suwancharoen et al., 2013). For the ELISA plates for common serovars in Thailand, the feasibility and effectiveness of preparing ELISA plates for the most common serovars in Thailand are supported by the need for accurate and rapid diagnosis of leptospirosis. The accuracy of ELISA for diagnosing human leptospirosis in Thailand has been still develop, with studies suggesting that the test may not be sufficiently accurate due to poor specificity and the potential for cross-reactivity with other tropical diseases (Desakorn et al., 2012). This underscores the importance of developing ELISA plates that are tailored to the prevalent local serovars, potentially improving diagnostic accuracy.

            Strategy for Regular Surveillance and Diagnostic Preparedness

            A comprehensive strategy for regular surveillance and diagnostic preparedness could include the following components:

  • Establishment of a Serovar Repository: Establishing a collection of local serovars would aid in continuous study and the creation of diagnostic tools, such as region-specific ELISA plates.
  • Periodic Review of ELISA Plate Serovar Composition: Consistently upgrading the ELISA plate composition using surveillance data will help maintain the relevance and effectiveness of the diagnostic instruments in identifying the most common serovars.
  • Enhanced Surveillance Programs: Developing advanced surveillance programs to track the distribution and prevalence of different types of leptospirosis would provide valuable information for public health policies and the improvement of diagnostic tools.
  • Research and Development: Stimulating research on recombinant antigen-based ELISA testing could enhance the sensitivity and specificity of diagnosing leptospirosis. (Chen et al., 2013).This approach could also facilitate the detection of antibodies against a broader range of serovars, including those prevalent in Thailand.
  • Collaboration and Data Sharing: Encouraging cooperation between public health organizations, research institutes, and the worldwide community would improve information exchange and the creation of successful diagnosis and surveillance methods.

            In conclusion, recognizing the significance of national surveillance for local serovars and creating ELISA plates specific to prevalent local serovars in Thailand is essential for enhancing the diagnosis and treatment of leptospirosis. Implementing a strategy that involves consistent monitoring, readiness for diagnosis, and the possible creation of a serovar repository could help direct future diagnostic methods in the area and lead to improved public health results.

References

Bajani, M. D., Ashford, D. A., Bragg, S. L., Woods, C. W., Aye, T., Spiegel, R. A., Plikaytis, B. D., Perkins, B. A., Phelan, M., Levett, P. N., & Weyant, R. S. (2003). Evaluation of four commercially available rapid serologic tests for diagnosis of leptospirosis. J Clin Microbiol, 41(2), 803-809. https://doi.org/10.1128/jcm.41.2.803-809.2003

Chen, H. W., Zhang, Z., Halsey, E. S., Guevara, C., Canal, E., Hall, E., Maves, R., Tilley, D. H., Kochel, T. J., & Ching, W. M. (2013). Detection of Leptospira-specific antibodies using a recombinant antigen-based enzyme-linked immunosorbent assay. Am J Trop Med Hyg, 89(6), 1088-1094. https://doi.org/10.4269/ajtmh.13-0041

Chirathaworn, C., Inwattana, R., Poovorawan, Y., & Suwancharoen, D. (2014). Interpretation of microscopic agglutination test for leptospirosis diagnosis and seroprevalence. Asian Pac J Trop Biomed, 4(Suppl 1), S162-164. https://doi.org/10.12980/apjtb.4.2014c580

Desakorn, V., Wuthiekanun, V., Thanachartwet, V., Sahassananda, D., Chierakul, W., Apiwattanaporn, A., Day, N. P., Limmathurotsakul, D., & Peacock, S. J. (2012). Accuracy of a commercial IgM ELISA for the diagnosis of human leptospirosis in Thailand. Am J Trop Med Hyg, 86(3), 524-527. https://doi.org/10.4269/ajtmh.2012.11-0423

Gómez-Morales, M., Selmi, M., Ludovisi, A., Amati, M., Fiorentino, E., Breviglieri, L., Poglayen, G., & Pozio, E. (2016). Hunting dogs as sentinel animals for monitoring infections with Trichinella spp. in wildlife. Parasites & Vectors, 9. https://doi.org/10.1186/s13071-016-1437-1

Gottschalk, M., Altman, E., Charland, N., De Lasalle, F., & Dubreuil, J. D. (1994). Evaluation of a saline boiled extract, capsular polysaccharides and long-chain lipopolysaccharides of Actinobacillus pleuropneumoniae serotype 1 as antigens for the serodiagnosis of swine pleuropneumonia. Vet Microbiol, 42(2-3), 91-104. https://doi.org/10.1016/0378-1135(94)90009-4

Jayasundara, D., Gamage, C., Senavirathna, I., Warnasekara, J., Matthias, M. A., Vinetz, J. M., & Agampodi, S. (2021). Optimizing the microscopic agglutination test (MAT) panel for the diagnosis of Leptospirosis in a low resource, hyper-endemic setting with varied microgeographic variation in reactivity. PLoS Negl Trop Dis, 15(7), e0009565. https://doi.org/10.1371/journal.pntd.0009565

Martin, E. A., Heseltine, J. C., & Creevy, K. E. (2022). The Evaluation of the Diagnostic Value of a PCR Assay When Compared to a Serologic Micro-Agglutination Test for Canine Leptospirosis. Front Vet Sci, 9, 815103. https://doi.org/10.3389/fvets.2022.815103

Ribotta, M. J., Higgins, R., Gottschalk, M., & Lallier, R. (2000). Development of an indirect enzyme-linked immunosorbent assay for the detection of leptospiral antibodies in dogs. Can J Vet Res, 64(1), 32-37.

Suwancharoen, D., Chaisakdanugull, Y., Thanapongtharm, W., & Yoshida, S. (2013). Serological survey of leptospirosis in livestock in Thailand. Epidemiol Infect, 141(11), 2269-2277. https://doi.org/10.1017/s0950268812002981

Woo, P. C., Leung, A. S., Lau, S. K., Chong, K. T., & Yuen, K. Y. (2001). Use of recombinant mitogillin for serodiagnosis of Aspergillus fumigatus-associated diseases. J Clin Microbiol, 39(12), 4598-4600. https://doi.org/10.1128/jcm.39.12.4598-4600.2001
